# Localization of infection in neonatal rhesus macaques after oral viral challenge

**Roslyn A. Taylor**[1], **Michael D. McRaven**[1], **Ann M. Carias**[1], **Meegan R. Anderson**[1],
**Edgar Matias**[1], **Mariluz Araínga**[2], **Edward J. Allen**[1], **Kenneth A. Rogers**[2],
**Sandeep Gupta**[3,4], **Viraj Kulkarni**[4], **Samir Lakhashe**[3,4], **Ramon Lorenzo-Redondo**[1,5],
**Yanique Thomas**[1], **Amanda Strickland**[4], **Francois J. Villinger**[2], **Ruth M. Ruprecht**[2,3,4],
**Thomas J. Hope**[1]*

1 Department of Cell and Developmental Biology, Northwestern University Feinberg School of Medicine, Chicago, Illinois, United States of America, 2 Department of Biology, New Iberia Research Center, University of Louisiana at Lafayette, Lafayette, Louisiana, United States of America, 3 Department of Microbiology, Immunology, and Molecular Genetics, University of Texas Health San Antonio, San Antonio, Texas, United States of America, 4 Disease Intervention and Prevention, Texas Biomedical Research Institute, San Antonio, Texas, United States of America, 5 Center for Pathogen Genomics and Microbial Evolution, Northwestern University Institute for Global Health, Chicago, Illinois, United States of America

* thope@northwestern.edu

**Data Availability Statement:** All relevant data are within the manuscript and its Supporting Information files.

## Abstract

Vertical transmission of human immunodeficiency virus (HIV) can occur *in utero*, during delivery, and through breastfeeding. We utilized Positron Emission Tomography (PET) imaging coupled with fluorescent microscopy of [64]Cu-labeled photoactivatable-GFP-HIV (PA-GFP-BaL) to determine how HIV virions distribute and localize in neonatal rhesus macaques two and four hours after oral viral challenge. Our results show that by four hours after oral viral exposure, HIV virions localize to and penetrate the rectal mucosa. We also used a dual viral challenge with a non-replicative viral vector and a replication competent SHIV-1157ipd3N4 to examine viral transduction and dissemination at 96 hours. Our data show that while SHIV-1157ipd3N4 infection can be found in the oral cavity and upper gastro-intestinal (GI) tract, the small and large intestine contained the largest number of infected cells. Moreover, we found that T cells were the biggest population of infected immune cells. Thus, thanks to these novel technologies, we are able to visualize and delineate of viral distribution and infection throughout the entire neonatal GI tract during acute viral infection.

## Author summary

Approximately 1.8 million children are currently living with human immunodeficiency virus (HIV). While mother-to-child HIV transmission can occur *in utero* and during delivery, it most commonly occurs through breastfeeding, creating the need to understand how the virus moves throughout the body and infects the infant once breast milk is consumed. Here, we used multiple imaging techniques and PCR to determine how HIV distributes throughout the gastrointestinal tract after oral viral exposure and in which tissues and cell types become acutely infected. We found that HIV rapidly spreads throughout and penetrates the entire gastrointestinal tract as early as four hours after exposure. We

**Funding:** This work was supported by the National Institutes of Health grants: NIH R01 DE023049 (RMR, TJH) and NIH 4T32AI00747620 (RAT) and 1 K01 OD026571-01 (AMC) (www.nih.gov). The funders did not play a role in the study design, data collection and analysis, decision to publish, or the preparation of the manuscript.

**Competing interests:** The authors have declared that no competing interests exist.

also found that the intestine contained the largest number of infected cells at 96 hours and that most cells infected were T cells. Our study shows that these imaging technologies allow for the examination of viral distribution and infection in a rhesus macaque model.

## Introduction

Mothers living with human immunodeficiency virus (HIV) and not on antiretroviral therapy have up to a 40% chance of passing HIV to their children [1]. Despite findings that suggest that exclusively breastfeeding infants can reduce HIV acquisition [2,3], breastfeeding remains one of the main routes through which vertical HIV transmission occurs [1,4]. This most likely occurs during the transition from breastfeeding to the introduction of solid foods [5] and possibly through the premastication of food by people living with HIV [6]. Regardless of their antiretroviral regimen, many women living with HIV need to breastfeed their infants due to limited resources in developing countries where the overall benefit of breastfeeding outweighs the risk for other life threatening infectious diseases, creating a great need to understand the mechanism of mother-to-child HIV transmission through oral viral exposure.

Non-human primates (NHP) provide a model of and an opportunity to investigate oral vertical HIV transmission *in vivo*. NHP models of oral HIV viral exposure results in rapid systemic infection using various methods of viral delivery as it has previously been shown that four-week-old neonatal rhesus macaques (RMs) have high viral blood titers after oral challenge with SIVmac251, which inversely correlated to survival [7]. Likewise, additional studies have proven similar; for example, direct application of SIVmac251 to tonsils and the check pouch resulted in systemic viral infection within seven days and two weeks, respectively [8,9]. Another study showed SIVmac251 DNA was concentrated in tissues of the head and neck, and systemic viral dissemination occurred four days after exposure when delivered dropwise into the mouth [10]. Lastly, recent reports also show that SIVmac251 RNA can be found in the brain and lungs 72 and 96 hours after challenge and SIV DNA is found throughout the gastro-intestinal (GI) tract 96 hours after oral challenge [11].

HIV infects and depletes CD4+ cells in mucosal tissues, resulting in a decrease of CD4+ T cells in the blood [12,13]. CD4+ T cells in neonates have higher rates of cell metabolism, proliferation and "activated" cell phenotypes [14,15] making them prime candidates for viral infection. Therefore, the rapid spread of viral infection in infants is most likely due to this immature highly metabolic immune system [16,17]. To demonstrate this CD4+ T cell vulnerability to acute viral infection, Amedee et al. (2018) [11], illustrated the presence of SIV RNA in tonsil and mesenteric lymph node T cells 72 hours after oral challenge [11]. Amedee et al. (2018) [11] also found that at 96 hours viral RNA was detected in ileum and/or colon. Furthermore, another study reported that bottle fed neonatal RMs resulted in a reduction in CD4 + counts in the blood during systemic infection a week after viral challenge [18]. Furthermore, neonates have a higher number of CD4+ T cells in various tissues compared to adults [19,20], which could contribute to viral spread. In adult RMs, recent studies have focused on identifying which CD4+ T helper (Th) cell subsets are highly susceptible to HIV/SIV infection, with an emphasis on Th17 cells [21]. It has been shown that Th17 cells are preferentially infected in adult RMs after acute viral exposure in the female reproductive tract [22]. Additionally, Th17 cells have been shown to play an important role in HIV infection in the adult human colon [23,24]. The potential role of Th17 cells has not yet been elucidated in neonates.

Despite the advances in understanding mother-to-child transmission, the exact sites of penetration of the various mucosal barriers of the alimentary canal to initiate transmission in breastfed infants remains unknown. This study utilizes three different viruses (or virus-like

particles) delivered in a bottle-fed neonatal RM model [25] to examine possible portals of viral penetration of mucosal barriers making them potential sites of virus acquisition and subsequent systemic infection after high-dose oral viral exposure. First, whole body Positron Emission Tomography and Computerized Tomography (PET/CT) imaging [26,27] was coupled with fluorescent microscopy of a photoactivatable-GFP HIV-BaL (PA-GFP-BaL) [28–30] to determine the distribution and localization of individual HIV virions two and four hours after viral challenge. Using PET/CT guided necropsy, we determined that HIV virions distribute to and penetrate the mucosal epithelium throughout the entire gastrointestinal (GI) tract, including the rectum, four hours after oral viral exposure. To study viral infection and viral cell targets, subsequent experiments used a dual viral challenge system, utilizing a non-replicative reporter virus [31] and a replicative SHIV-1157ipd3N4 [32]. Using our non-replicative reporter virus, we found the tongue may be a main site of viral transduction. Additionally, in congruence with previous findings [9–11], we found that the entire GI tract is susceptible to SHIV-1157ipd3N4 infection 96 hours after oral challenge. Likewise, the small intestine was identified as the tissue that held the biggest foci of infection by fluorescent deconvolution microscopy, with T cells encompassing the largest infected population of cells. These findings can provide mechanistic insight and increase our understanding of mother-to-child transmission after viral exposure.

## Results

### PET/CT imaging illustrates that HIV virions distribute throughout the GI tract four hours after oral viral exposure

The technologies of PET/CT imaging, photoactivatable HIV (PA-GFP-BaL), and fluorescent deconvolution microscopy [28–30] were combined to determine where virus distributes throughout the body after oral exposure. PET/CT imaging allows for the *in vivo* tracking of virions over time after challenge through radiolabeling PA-GFP-BaL with a radioactive isotope of copper, $^{64}$Cu (**Fig 1A**). During tissue processing, tissues are cut into 1cm pieces, frozen into blocks with optimal cutting temperature medium (OCT), and each block is scanned to identify individual blocks with positive radioactivity, thereby increasing our chances of finding HIV virions by microscopy (**Fig 1B and 1C** [30]). These experiments occurred in pairs, with one two-hour animal and one four-hour animal for each experimental day. The same viral stock was used in all four animals.

Two hours after oral exposure of PA-GFP-BaL-$^{64}$Cu, PET scans revealed radioactivity throughout the GI tract up to the transverse colon in both two-hour animals. Four hours after oral exposure, radioactivity was observed throughout the GI tract to the colon including the length of the rectum (**Fig 2A–2D**). In one of the four-hour animals, PET signal was not found past the descending colon, which was most likely due to a bubble of trapped gas that was observed upon colon resection (**Fig 2C**). PET imaging of individual tissue blocks confirmed our findings from the whole-body PET imaging in all four animals (**Fig 1B**); despite the gas obstruction observed in one animal, individual radioactive blocks up to and including the transverse colon were found in both two-hour animals and throughout the length of the GI to the rectum in both four-hour animals (**Fig 2C**).

### HIV virions distribute throughout the entire length of the GI tract four hours after oral viral exposure

Next, fluorescent deconvolution microscopy was performed to identify the total number of individual HIV virions (non-penetrating and penetrating) in individual tissue blocks that had

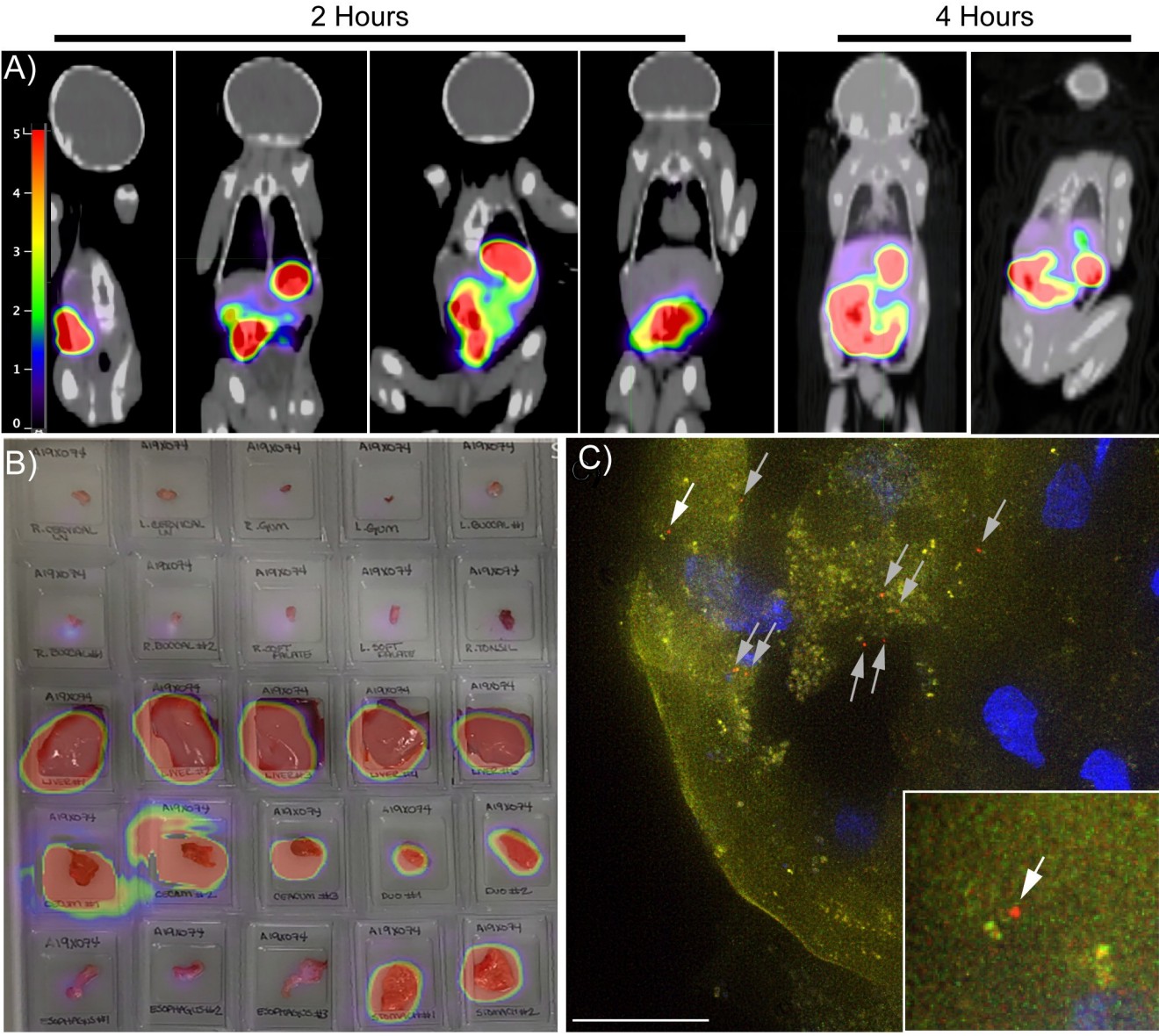

**Fig 1. PET imaging follows distribution of PA-BaL-⁶⁴Cu after oral viral challenge.** Four animals were orally challenged with PA-GFP-BaL-⁶⁴Cu. Two animals were sacrificed two hours post challenge and another two four hours post challenge, and the oral cavity and entire GI tract removed in one piece. The tissues were cut into pieces, frozen, cryosectioned, and prepared for fluorescent microscopy. **A,B)** Representative PET images of neonates after oral viral challenge with PA-GFP-BaL-⁶⁴Cu. Scale in Standard Uptake Value (SUV) **(A)** Whole body PET images at two and four hours post-oral challenge. **B)** PET image overlaid on photograph of 25 individual tissue blocks from oral mucosa and GI tract four hours post-challenge. **C)** Representative fluorescent microscopy image showing individual HIV virions (red puncta indicated by arrows) penetrating the tongue of an animal that received PA-BaL-⁶⁴Cu at two hours post-challenge. Green–pre-activation, Red–post-activation (virion), Blue–Hoechst. White arrow indicates virion shown in inset; grey arrows indicate other virions in the micrograph.

strong radioactivity by PET imaging (**Fig 1C**). Penetrating virions were defined as being one micron from the epithelial surface. Because virions were not found in every tissue of the GI tract in all animals, the GI tract was dichotomized into upper (esophagus and stomach) and lower (small intestine and colon) sections.

The focus of these analyses was to explore statistically significant differences in the anatomical distribution of virions and their characteristics, not the differences between animals. The

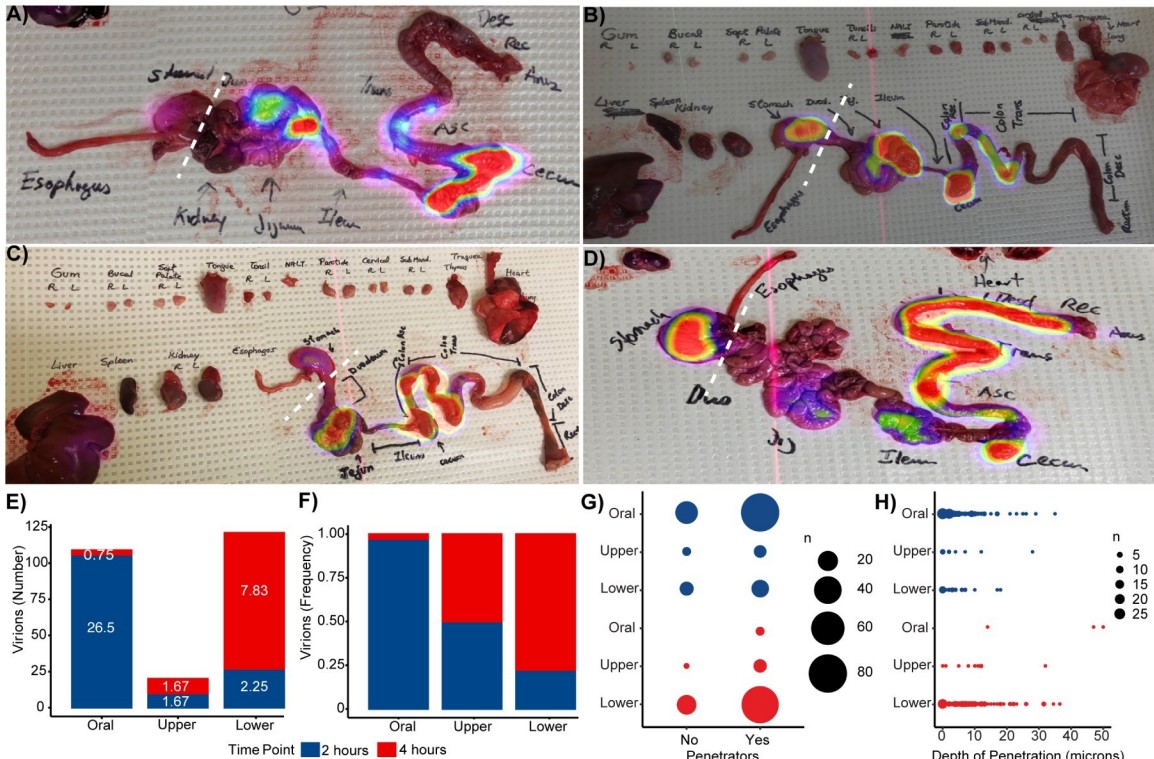

**Fig 2. HIV virions distribute throughout and penetrate the mucosal epithelium of the entire GI tract four hours post-oral challenge.** Whole GI tracts were excised and imaged by PET at two- and four-hours following necropsy. PET images overlaid on photographs allow for visualization of radioactivity throughout the GI tract at two hours **(A-B)** and four hours **(C-D)** post challenge. White dotted lines depict where the GI tracts were dichotomized into upper (esophagus and stomach) and lower (small and large intestines) regions. Quantification of virions and viral penetration in the oral cavity, upper GI tract, and lower GI tract mucosa two and four hours after oral viral challenge. **E)** Total number of virions counted in images. Means of the virion count per group shown on stacked bars in white numbers. **F)** Frequency of total virions counted in each specific anatomic site; frequencies obtained by combining the virions in both animals at each time point, respectively. **G)** Quantification of the number of penetrating virions. Circle area represents the number of virions. **H)** Virion penetration depth in microns. Each virion depth was truncated to integer numbers for representation purposes. Circle area represents the number of virions with the same depth value. Blue: two hours, Red: four hours.

number of animals in the study was insufficient for comparison between animals. With this objective, due to our low number of animals, we performed an exhaustive imaging of tissues to achieve an extensive sampling of virions in each tissue of the animals analyzed. In the different analyses we used different mixed-effect models, as described in the methods, always using each animal as random effect to control for the lack of independence of the data sampled within each animal. Therefore, the results reflect and account for this fact. Afterwards, within the different models used and depending on the nature of the data analyzed, we performed all possible contrasts between every location within such fitted models to analyze statistically significant differences in the number of virions and/or their characteristics in each location. In order to do so, we used the Benjamini-Hochberg (BH) Procedure, as described in the methods, which controls for False Discovery Rate (FDR). When using BH, the sample size is relevant in expecting or not significant differences, as it is just a method to avoid Type I errors. Moreover, the power of our analysis is derived from the extensive sampling of virions in different tissues through multiple images from the animals.

Although the distance to which HIV virions distributed after oral challenge extended to the transverse colon, the majority of the virions were located in the oral cavity (buccal, tongue, and tonsil; mean = 26.5 virions, frequency = 0.97; A19X042 = 101 virions, A19X073 = 5

virions) in animals necropsied two hours post-challenge. We also observed significantly fewer numbers (FDR<0.05) and a lower frequency of virions both in the upper (mean = 1.67 virions, frequency = 0.50; A19X042 = 6 virions, A19X073 = 4 virions) and lower (mean = 2.25 virions, frequency = 0.22; A19X042 = 24 virions, A19X073 = 0 virions) GI tract compared to the oral cavity two hours after viral exposure (**Fig 2E and 2F**). However, in the four-hour animals, fewer virions were found in the oral cavity (mean = 0.75 virions, frequency = 0.03; A19X038 = 0 virions, A19X074 = 3 virions). The highest number of the virions found in the four-hour animals was in the lower GI tract (mean = 7.83 virions, frequency = 0.78; A19X038 = 58 virions, A19X074 = 36 virions **Fig 2E and 2F**). The same number and frequency of virions were found in the upper GI tract (mean = 1.67 virions, frequency = 0.50; A19X038 = 0 virions, A19X074 = 9 virions) in the four-hour animals as the two-hour animals.

Similar results were found when specifically examining the number of virions that penetrated into the mucosa (**Fig 2G**). In the two-hour animals, the largest number of penetrating virions were found in the oral cavity (sum of virions in both two-hour animals: non-penetrating = 26 virions, penetrating = 80 virions) with statistically significant differences observed both between this cavity and the upper (non-penetrating = 3, penetrating = 7) and lower GT tract (non-penetrating = 9, penetrating = 15; FDR<0.05). Likewise, in the four-hour animals, the majority of penetrating virions were found in the lower GI tract (non-penetrating = 19 virions, penetrating = 75 virions), although among the very few virions found in the oral cavity, all of them were penetrating virions (penetrating = 3). In the upper GI tract, few virions were found at four fours (non-penetrating = 1, penetrating = 8) There was no difference in penetration depth between location of virion or time point after oral challenge (**Fig 2H**) with the exception of the few virions found in the oral cavity at four hours that proportionally were also significantly deeper penetrators. These data, combined, demonstrate the validity of PET/CT to identify areas of virus accumulation along with illustrating that HIV virions can distribute throughout the entire GI tract four hours after oral viral exposure in neonates.

## Validating our dual challenge oral transmission model

While PET/CT and PA-GFP-BaL-$^{64}$Cu experiments allow for the visualization of virion distribution and penetration into mucosal tissues, these technologies do not allow for the examination of viral infection. Therefore, to study viral infection, subsequent experiments orally challenged neonatal RM with LICh, a non-replicative reporter virus [31], and a replicative SHIV-1157ipd3N4 [32]. Our lab has previously shown that we can locate and identify cells that were first transduced by the challenge inoculum through a viral vector, LICh [31]. Due to the lack of accessory genes in LICh, the replication process gets halted after one round of viral integration, thus resulting in viral transduction. In a proof-of-principle experiment, one animal was orally challenged with 8mL of LICh alone and then sacrificed at 96 hours after challenge to validate whether LICh can be used to identify sites of viral transduction after oral challenge. Tissues from the GI tract were extracted and analyzed by IVIS. We were able to identify luciferase signal on IVIS in the esophagus, cervical lymph nodes, and stomach of neonatal RMs at 96 hours after oral exposure (**S1 Table**).

## Identifying tissues susceptible to viral transduction and infection

Since we were able to observe luminescence by IVIS after oral challenge, we proceeded to use a dual viral challenge model for the remaining experiments. We have previously shown that LICh can be used as a guide to locate areas of replicative foci of infection, as LICh disseminates throughout the body similar to replicative viruses [22,31]. Because LICh does not replicate, an SIV containing an R5-tropic HIV Clade C *env* genes, SHIV-1157ipd3N4, was used to study

viral replication. The use of SHIV-1157ipd3N4, which was isolated from a pediatric HIV patient, allows for a translationally relevant model for investigating lentiviral pathogenesis of subclade C envelope, which is most dominant HIV subclade in sub-Saharan Africa [32]. Because LICh does not contain accessory genes, we could distinguish SHIV-1157ipd3N4 infected cells from LICh transduced cells by examining replicative viral infection by gag DNA and protein. To identify tissues that are initially susceptible to viral transduction and infection, genomic DNA was extracted and performed nested PCR to target mCherry and gag DNA. In a proof-of-principle experiment, one neonatal RM (RM13) was challenged with four feedings of 2mL of LICh and a repeated low dose of SHIV-1157ipd3N4 (see Experimental Methods) and necropsied shortly after two days of viral challenge to examine early transmission events. Results from the low dose challenge and 53-hour time point showed less transduction and infection, as predicted. At 53 hours post-oral challenge, we found mCherry DNA in the top of the tongue (Table 1). Gag DNA was found in the cervical lymph nodes, stomach, and small intestine at 53 hours post-challenge.

For the next set of experiments, eight neonatal RMs were orally challenged with a repeated high dose challenge of SHIV-1157ipd3N4 and one 8mL bolus of LICh, which was given during the last bottle feeding, and the time between challenge and necropsy was extended to 96 hours prior to nested PCR and fluorescent microscopy being performed, increasing the potential of finding infected cells and of understanding the mechanism of viral transduction and infection in our bottle-feeding model. While luciferase activity was previously observed in the proof-of-principle study, when this group of eight neonates were examined by IVIS, very little luciferase activity was seen among all the animals; therefore, we performed nested PCR on all processed tissues. Nested PCR revealed mCherry DNA in the tongue, stomach, and small intestine in RM10 and in the tongue of RM17. At 96 hours, SHIV-1157ipd3N4 viral dissemination was more widespread. Gag DNA was found in the cheek, tonsil, soft palate, esophagus, mesentery, small intestine, large intestine, liver, and spleen of RM10. Gag DNA was also in the cheek, tongue, tonsil, the transformation zone to the stomach from the esophagus, small intestine, large intestine, spleen, and cervical lymph nodes of RM17 (Table 1). When performing nested PCR, large amounts of tissues are sectioned for DNA isolation. Therefore, to minimize tissue sample depletion, future experiments primarily focused on fluorescent microscopy to identify viral transduction instead of utilizing nested PCR to identify infected tissues.

## LICh viral vector is not efficient in identifying transduced cells by fluorescent microscopy after oral viral challenge

Tissue sections were then examined by fluorescent deconvolution microscopy to identify individual LICh transduced cells at 96 hours (Table 2). After a careful and thorough examination

**Table 1. Distribution of LICH transduction and SHIV-1157ipd3N4 DNA throughout the GI tract after oral challenge.**

| Animal | Challenge | Tongue | Cheek | Soft Palate | Tonsil | Cervical Lymph Nodes | Esophagus | Stomach | Small Intestine | Large Intestine | Spleen | Liver |
|---|---|---|---|---|---|---|---|---|---|---|---|---|
| RM13 | Low Dose 53 hrs | + | | | | # | # | # | # | | | |
| RM10 | High Dose 96 hrs | + | # | # | # | | # | + | +# | # | # | # |
| RM17 | High Dose 96 hrs | + # | # | | # | # | # | # | # | # | # | |

LICh viral vector mainly found in the tongue after oral challenge. SHIV-1157ipd3N4 viral dissemination found throughout the GI tract after oral challenge. + indicates mCherry DNA found; # indicates gag DNA found.

**Table 2. Localization of LICh transduced cells at 96 hours post oral challenge by microscopy.**

| Animal | Tongue | Trachea | Stomach |
|--------|--------|---------|---------|
| RM10 | | | |
| RM17 | | | |
| RM22 | 1 | | |
| RM23 | | 1 | |
| RM25 | | | 5 |
| RM26 | | | |
| RM27 | | | |
| RM28 | | | |

Number of LICh transduced cells found in the tongue, trachea, and stomach 96 hours after oral challenge by microscopy. mCherry signal was validated by spectral imaging.

of all tissues, minimal mCherry+ cells were found in only three of the neonates (**Table 2**). However, although we were able to detect and verify mCherry+ cells in a few animals, these cells proved difficult to widely identify due to the high autofluorescent background in the neonatal tissues. Therefore, tissue sections that contained luciferase activity by IVIS were also stained with antibodies for luciferase. Unfortunately, despite tissues showing luciferase activity by IVIS, positive luciferase signal was not detected in any of the neonatal tissues. Although, we previously used a LICh viral vector in a model of SIV infection in the female reproductive tract to identify sites of viral transduction [31], these data suggest that in our oral viral challenge model, LICh technology may not be as efficient. Therefore, defining SHIV-1157ipd3N4 viral infection, replication, and dissemination in the oral and gut mucosa was prioritized.

## Most infected cells are found in the small intestine after oral viral challenge

To determine which cell types are susceptible to replicative viral infection after oral exposure, tissues were examined for evidence of SHIV-1157ipd3N4 infection using fluorescent deconvolution microscopy (**Table 3**). Tissue sections from the oral cavity and GI tracts of each animal were stained with antibodies directed toward SIV Gag (clone AG3) to identify SHIV-1157ipd3N4 gag, as well as CD3 and CCR6 to phenotype infected immune cell subsets (**Fig 3, Table 3**). Five panels consisting of three 40x by five 40x images were taken for phenotype analysis. As we have previously shown, these markers allow us to identify the target cells as Th17, T cells, immature dendritic cells (iDCs), and other. In the tongues of neonatal RMs, we found very small foci of infection; five of the eight animals examined had SHIV-1157ipd3N4 infected

**Table 3. Localization of SHIV-1157ipd3N4 infected cells 96 hours after oral challenge by microscopy.**

| Animal | Tongue | Tonsil | Esophagus | Stomach | Small Intestine | Large Intestine |
|--------|--------|--------|-----------|---------|-----------------|-----------------|
| RM10 | 7 | 3 | 1 | 3 | 137 | 203 |
| RM17 | 0 | 51 | 0 | 38 | 400 | 292 |
| RM22 | 0 | 1 | 0 | 20 | 216 | 60 |
| RM23 | 1 | 5 | 36 | 13 | 980 | 142 |
| RM25 | 8 | 0 | 0 | 20 | 56 | 95 |
| RM26 | 21 | 9 | 2 | 17 | 194 | 40 |
| RM27 | 10 | 39 | 1 | 48 | 506 | 204 |
| RM28 | 0 | 0 | 0 | 192 | 636 | 223 |

Number of SHIV-1157ipd3N4 infected cells found throughout the oral mucosa and GI tract after oral challenge.

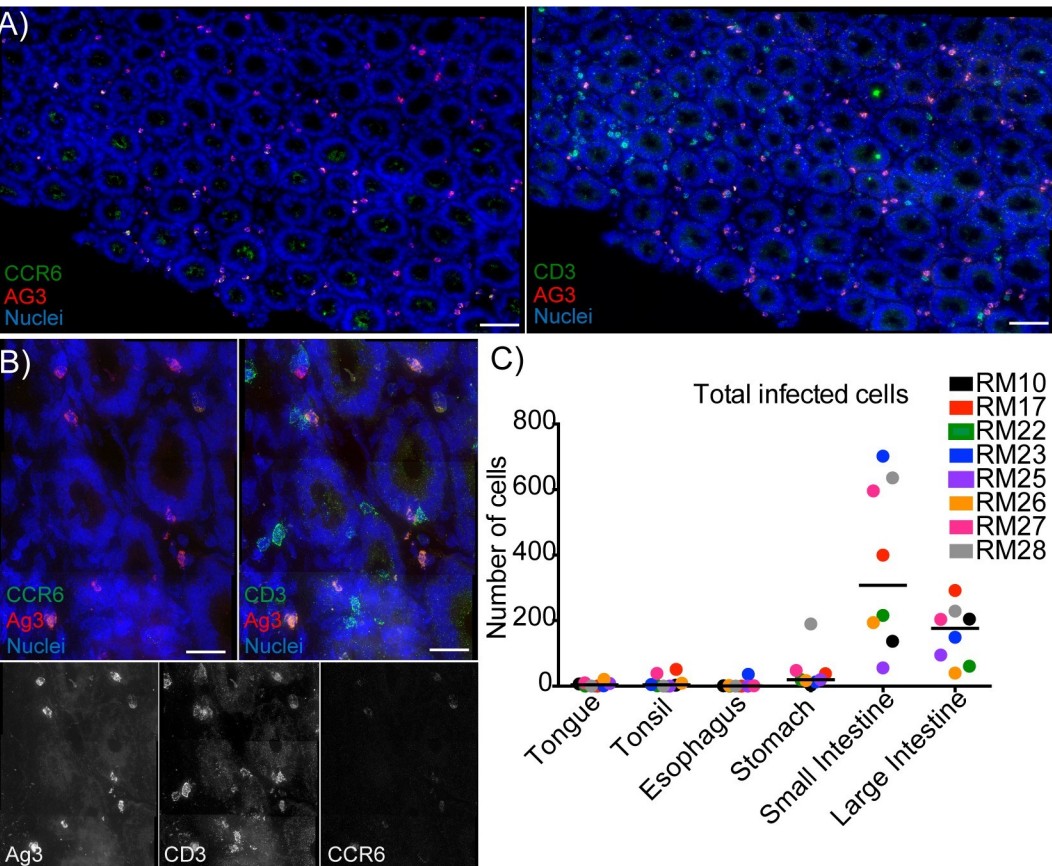

**Fig 3. The majority of SHIV-1157ipd3N4 infected cells are found in the small intestine 96 hours after oral challenge.**
Fluorescent microscopy and quantification of SHIV-1157ipd3N4 infected cells from eight animals that were sacrificed at 96 hours after dual viral oral challenge. Tissues were dissected, analyzed by IVIS, frozen, and tissue blocks with high luciferase activity were cryosectioned. Slides were stained for CD3, CCR6, AG3, and Hoechst. Representative fluorescent microscopy images. **A-B)** Cryosections of the small intestine showing SHIV-1157ipd3N4 infected cells **A)** 40X panels, scale bars 60 microns **B)** 100x panels, scale bars 20 microns **C)** Quantification of total number of SHIV-1157ipd3N4 infected cells by tissue type. Each dot represents the total number of cells found in five 40x panels in one individual animal.

cells (5.86 ± 7.36 cells, median = 4, **Fig 3C**). Similar to what was observed in the tongue, the tonsils contained a small number of SHIV-1157ipd3N4 infected cells in six of the animals (13.5 ± 19.93 cells, median = 4), corroborating previous studies illustrating tonsils as a site of potential portal of transmission after oral viral challenge [8,10,18,33]. The fewest number of infected SHIV-1157ipd3N4 cells in terms of number of animals and quantity of cells were in the esophagus (5 ± 12.55 cells, median = 0.5). While SHIV-1157ipd3N4 infected cells were detected in the tongue, tonsils, and esophagus, it was not a common event. In contrast, the stomach contained larger foci of infection compared to those found in the oral cavity and esophagus; all eight RMs had infected cells in the stomach (43.63 ± 60.82 cells, median = 20).

Previously, it has been shown that the intestines are a site of viral expansion in neonates after intravenous infection [34]; therefore, we examined the small intestine, large intestine, and their draining mesenteric lymph nodes for foci of SHIV-1157ipd3N4 infection to see if we could identify similar after oral challenge. All eight RMs in the study had SHIV-1157ipd3N4 infected cells throughout the small and large intestine. The small intestine contained the largest foci of infection in all eight neonates (367.1 ± 250.8 cells, median = 308, **Fig 3A and 3B**). Overall, the large intestine had the second largest foci of infection in the neonates (159.4 ± 88.39,

median = 176.5, infected cells, **Figs 3C and 4A**). Unfortunately, we were only able to obtain the mesenteric lymph nodes from three of the eight animals; however, we found SHIV-1157ipd3N4 infected cells in the mesenteric lymph nodes of two of these three animals (**Table 3**). These data suggest that viral replication can be found throughout the GI tract, all the way to the distal large intestines and the corresponding mesenteric lymph nodes after oral viral challenge.

## The majority of SHIV-1157ipd3N4 infected cells throughout the gut are T cells at 96 hours after oral viral challenge

It has previously been shown that neonates have CD4+ T cells of a memory phenotype that proliferate at high rates in mucosal tissues, which may make these T cells the primary target of HIV/SIV infection [34]. Immature dendritic cells (DC) have been suggested to be initial targets and mediators of HIV/SIV [35–37] infection. Our lab has previously shown that Th17 cells are preferentially infected after vaginal SIV challenge [22]. Taking all these results together, we stained our tissue sections for T cell marker CD3 and the chemokine receptor CCR6 to investigate which cell types are infected after oral challenge. Infected cells were phenotyped into the following groups: Th17 T cells (CD3+ CCR6+ AG3+), CCR6- CD3+ T cells (CD3+ AG3+), Immature DC (CCR6+ AG3+), and other (AG3+). Due to the small number of SHIV-1157ipd3N4 infected cells found in the tongue, tonsil, and esophagus, our phenotyping analysis focused on the stomach, small intestine, and large intestine. In the stomach, the majority of infected cells were other T cell subsets when examining all eight neonates combined (n = 242 cells, 68.95%, **Fig 4B**). Overall, Th17 T cells were the second largest infected population in the stomach (n = 78 cells, 22.22%). Individually, Th17 T cells made up the largest group of infected cells in three of the RMs (RM17, RM22, RM26), while other T cell subsets were the largest group of infected cells in 4 of the RMs (RM23, RM25, RM27, RM28, **S1 Fig**). Other cell types made up 6.55% of total infected cells (n = 23 cells) and immature DC were 2.28% of infected cells (n = 8 cells) in the stomach. One neonate, RM10, had very few infected cells (n = 3), however 33.33% of infected cells of Th17 cells, (n = 1 cell) CCR6- CD3+ T cells subsets (n = 1 cell), and immature DC (n = 1 cell, **S1 Fig**). The largest infected population of cells in the small intestine was CCR6- CD3+ T cells in all eight neonates (n = 2468 cells, 76.74%, **Figs 4B and S1**). The second largest infected population found in the small intestine was Th17 T cells (n = 655 cells, 20.37%). Similar to the stomach, few other cell types (n = 86 cells, 2.67%) and immature DC (n = 7 cells, 0.22%) were infected in the small intestine. Other T cell subsets were marginally infected with the most in the large intestine (n = 651 cells, 51.06%). For

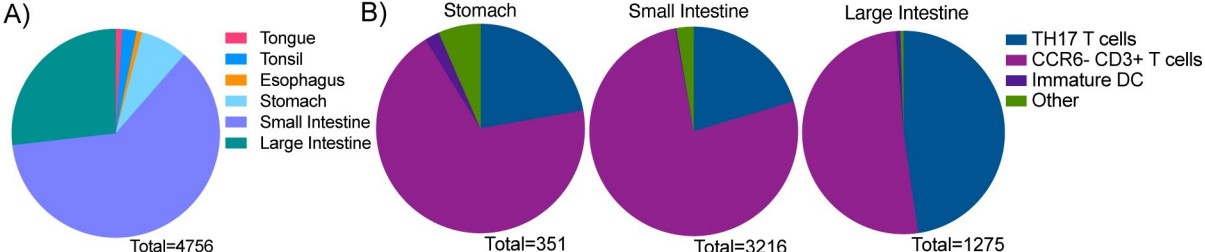

**Fig 4. The majority of SHIV-1157ipd3N4 infected cells are T cells at 96 hours after oral challenge.** Quantification of SHIV-1157ipd3N4 infected cells found in the oral cavity and GI tract of animals examined in Figs 3 and 4. Graphs depict the percentage of infected cell types identified by fluorescent microscopy as parts of a whole. Total cell counts were taken in five 40x panels in every animal. **A)** Total number of infected cells in all eight neonatal RMs in tongue, tonsil, esophagus, stomach, small intestine, and large intestine. **B)** Total number of infected cell phenotypes in all eight RMs in the stomach, small intestine, and large intestine. Infected cell types were categorized as five cell types: T cells (CD3+), TH17 T cells (CD3+, CCR6+), CCR6- CD3+ T cells, Immature DCs (CD3-, CCR6+), Other (CD3-, CCR6-).

example, the large intestine was the site of the greatest infection in Th17 T cells (n = 608 cells, 47.69%, **Figs 4B** and **S1**); the majority of infected cells in half of the neonates were Th17 T cells (RM10, RM17, RM26, RM27), while other T cell subsets were in the remaining four (RM22, RM23, RM25, RM28). Very few infected immature DC (n = 10 cells, 0.78%) and other cell types (n = 6 cells, 0.47%) were found in the large intestine. Overall, our data suggest that T cell subsets that are not Th17 are the greatest target of infection in our oral transition model, unlike our previous findings after vaginal challenge [22]. Among all the animals and tissue types analyzed, very few immature DC and other cell types were infected.

## Discussion

Previous studies using intravenous inoculation of infant rhesus macaques have provided great insights into the susceptibility of neonatal RMs to SIV infection [38,39] and CD4+ depletion in the gut after challenge [14,15,34]. However, breastfeeding remains the most prevalent route for vertical HIV transmission worldwide [1]. This creates a great need to study oral transmission in neonatal animal models. Here we show that, in a bottle-fed model, HIV virions rapidly distribute throughout the GI tract and intact particles found as far as the rectum four hours after oral exposure (**Figs 1** and **2**). We also show that the greatest number of SHIV-1157ipd3N4 infected cells is in the small intestine and that T cells are primarily infected 96 hours after viral exposure (**Figs 3** and **4**). Our findings are in congruence with the findings of previous studies, including Amedee et.al. who examined locations of SIV RNA and DNA over 96 hours, which showed that the ileum was the location with the second highest concentration of SIV RNA. However, in this study, it was reported that the location with the highest SIV RNA and DNA were the retropharyngeal lymph nodes [11]. It is important to note that this study examined SIV RNA and DNA only, which could possibly explain the differences observed in our study; likewise, we did not examine the retropharyngeal lymph nodes, which could be of interest in a future study. The observation that viral RNA could be detected in the iIleum and colon 96 hours after oral challenge is consistent with the abundant foci of infection detected in the gut 96 hours after challenge.

To determine how HIV virions distribute throughout the body immediately after oral exposure, we radiolabeled a photoactivatable, fluorescently tagged HIV and performed *in vivo* PET imaging coupled with deconvolution fluorescent microscopy (**Figs 1** and **2**). While these PET/CT experiments are not studying viral infection, these technologies provide insight into how HIV disseminates throughout the body, and where in the oral mucosa and GI tract the virus enters the mucosa after oral viral exposure. We found that two hours after oral challenge, HIV virions mainly localize in the oral cavity, but then further disseminate to the transverse colon. Four hours after challenge, HIV virions are found throughout the small and large intestine; in both four-hour animals, many virions were also found within the rectum. These experiments revealed that HIV distributes throughout the GI tract rapidly after oral challenge. In a matter of two hours, the majority of total virions and penetrating virions changed from the oral cavity to the lower GI tract consistent with virion penetration of epithelial barriers happening in a wave followed by lysis or turnover because the signal was lost at the two-hour timepoint (**Fig 2**). It has been reported that the gastric juice of neonates has a relatively neutral pH compared to adults, ranging mostly between 7.5 and 8.5, which could explain how the virus was able to survive and pass through the stomach of the neonatal RMs [40].

Previously, it has been suggested that the tonsils may be a portal of viral access to susceptible immune cells after oral exposure [8,10]; however, these studies directly applied virus to the tonsils. We found very few SHIV-1157ipd3N4 infected cells in these tissues 96 hours in our bottle-fed model of oral viral exposure (**Figs 3C** and **4A**). Although these data do not discount

the tonsils as a potential portal of transmission, it suggests that the tonsils and other secondary lymphoid aggregates of the oral cavity may not be the primary or major portal of neonatal acquisition through breast feeding as suggested by some models [8,10]. It has previously been shown that only 5% of CD4+ T cells in human tonsils are productively infected in an *ex vivo* model using an X4-tropic HIV [41]. Instead, the cell death that is observed in the tonsils is a result of apoptosis of uninfected, bystander CD4+ T cells [41]; this same phenomenon has been observed in lymph nodes from children living with HIV and SIV infected RMs [42]. It is notable that the tonsils are more similar to lymph nodes than the mucosal environment of the small and large intestine. Importantly, most of the T cells in the tonsil do not express the chemokine receptor CCR5, which is a required co-receptor which in combination with CD4 is required for the virion to functionally fuse with the target cell membrane. CCR5 is enriched at mucosal sites; low levels of CCR5 are observed in lymphatic tissue such as lymph nodes and tonsil. The paucity of susceptible T cells (CCR5+) in the tonsils is the likely reason that we found limited numbers of SHIV-1157ipd3N4 infected cells in the tonsil and oral cavity overall. In contrast, the gut contains a large number of potential SHIV-1157ipd3N4 target cells with CCR5+, CD4+ T cells being the majority population in the immune cell rich mucosal environment of the intestine. This makes the gut a rich environment for the virus to replicate which is reflected in the presented data, especially the small intestine (**Figs 3 and 4**). Considering our observations, our data suggests that the gut may represent the major portal of viral transmission after oral exposure by breastfeeding. Consistent with this possibility, the distribution of viral particles in a liquid, oral challenge rapidly reaches the stomach and intestines where there will be ample opportunity for the virus to reach and penetrate the luminal barrier of the gut. In contrast, the fluid in the inoculum will rapidly flow over the surface of the oral cavity giving less of an opportunity to penetrate the epithelial barriers of the oral cavity which is target cell poor environment. While we found the majority of SHIV-1157ipd3N4 cells in the small and large intestine, it is important to note that we cannot determine with our methods if these large foci of infection represent the initial sites of infection at the portal of transmission or a location rich in target cells that can foster high levels of viral replication after initial expansion and replication in the oral cavity. It seems the large number of expanding, substantial infectious foci in the gut is most likely initiated in the gut considering the short four-day period between drinking virus containing fluid and necropsy. The efficient movement of the virus containing fluid we documented throughout the alimentary canal four hours after drinking and our ability to identify intact viral particles penetrating the intestinal mucosa likewise support a model where oral exposure of virus containing fluid can utilize the intestine as a portal of transmission.

Another retrovirus known to be milk transmitted is the mouse mammary tumor virus (MMTV) that disseminates through the GI tract and directly infects immune cells in the Peyer's patches of the gut, suggesting that other breast milk-transmitted retroviruses may also have this capability [43,44]. Although current models of MMTV acquisition suggest that this process takes place through cell-associated virus (infected cells in milk) which could withstand the acidic pH of stomach acid better than cell-free virus [45]. However, our observations clearly demonstrate the viral particle can reach the intestinal compartment intact, without being cell associated, agreeing with previous results from Baba et. al. which demonstrated that cell-free SIV is transmitted orally in neonatal rhesus macaques [38]. Providing the challenge virus in formula may have played a role in buffering the mucosal environment slowing virion lysis and degradation. Therefore, our PET/CT and microscopy data, taken together with the knowledge that the gut is a target rich environment for viral infection is consistent with a model where the SHIV-1157ipd3N4 infected cells we found in the intestines originated from virus in the challenge inoculum utilizing the gut as the primary portal of transmission.

However, due to the replicative properties of SHIV-1157ipd3N4 we cannot rule out the possibility that the infected cells found in the small and large intestine originate from virus that had disseminated from the portal of transmission in the oral cavity. But assuming a 24-hour replication cycle for SIV, it seems unlikely that 96 hours is enough time to generate the large foci of infected cell we observe in gut mucosa. It is notable that we observe similar size foci of infection at 96 hours after vaginal or rectal challenge. It is notable that Amedee et al. (2018) [11] also detected gut infection by RNA PCR 96 hours after oral challenge, but interpreted the gut signal to be a consequence of virus dissemination from the oral cavity because viremia could also be detectable at the 96 hour timepoint. However, our observation of intact viral particles penetrating the gut mucosal barriers after oral challenges increases the potential that oral exposure can initiate infection in the gut. Future experiments are required to determine the origin of the foci of infection we observed.

Our lab has previously shown that the reporter viral vector, LICh, allows us to locate tissues and cells that were transduced by the challenge inoculum through bioluminescence, nested PCR and fluorescent microscopy [22,31]. This technology provides us with the ability to identify cell types that are vulnerable to becoming the first cell infected in different models of HIV transmission. In our model of oral HIV transmission, we were able to use LICh to identify sites of viral transduction through the presence of mCherry DNA by PCR (**Table 1**). However, locating mCherry+ cells by microscopy was arduous (**Table 2**). It may be possible that, while our viral vector works well in rectal and vaginal infection models, this technology is not well suited for oral challenge. Visualizing LICh is optimal with a focal area of transduction to produce a sufficient density of photons for efficient detection. Therefore, a target poor environment like the tonsils may not be able to achieve sufficient photon flux for detection. Challenge with LICh without mixing with the SHIV was also inefficiently detected. Likewise, the particles will be greatly diluted and at low concentration to transduce cells at a density conducive to the detection of luciferase activity. Because of inefficient photons to guide our efforts to identify pieces of tissue containing foci of transduction, we decided to forgo searching for LICh transduction in subsequent animals (RM23, RM25, RM26, RM27, RM28) and proceed to identify SHIV-1157ipd3N4 infected cells. The large replicative capacity of the virus, and infectious clone engineered to contain NF-KB sites in the long terminal repeats [32], facilitated our ability to identify pieces of tissue containing foci of viral replication random screening.

Models of mother-to-child transmission that have used oral routes of inoculation utilized either dripping virus directly onto the tonsils [9,10,18,46], slowly dripping cell-free virus solution onto the back of the tongue [38] or bottle-feeding methods [18,47]. To mimic how vertical transmission occurs in humans via breastfeeding, we mixed challenge inoculums into formula, which was then bottle fed [25]. Currently, the exact dose of HIV in each exposure during breastfeeding in humans is unknown [1]. It has been shown that human breast milk viral load can range widely from a hundred copies per milliliter to millions of copies per milliliter. These studies have also shown that a high viral load in breast milk correlates to HIV transmission in human infants [48–50]. Our initial experiments were based on repeated low dose viral challenges as other studies have previously reported [18,46]. We found SHIV-1157ipd3N4 DNA (**Table 1**), however, viral dissemination was not as widespread as we predicted using the repeated low challenge dosing. Therefore, to better understand the mechanisms of viral dissemination after oral challenge, we increased the dose of virus in our repeated challenges to super-physiological. From these changes in experimental design, along with an increase in time point to day four post viral challenge, infected cells could be detected easily throughout the GI tract (**Figs 3, 4 and S1 and Table 3**). It is important to note that in our model of oral viral challenge, the viral inoculum was diluted for delivery in Pedialyte (see Experimental Methods), which is not physiologically representative of breast milk. Breast milk components

have been shown to serve as antimicrobial and anti-inflammatory agents [51]. Lactoferrin has antiviral properties that can inhibit HIV-induced cytopathic effect [52,53]. Oligosaccharides found in breast milk have been shown to be effective inhibitors of HIV. Breast milk also contains an abundance of maternal antibodies that could protect infants against infection [51]. These protective components most likely are the reason not every breastfed baby acquires HIV when orally exposed. The lack of these protective components in our model could partially explain why we found infected cells in the GI mucosa in every animal in our study.

Neonates have an immature innate and adaptive immune system that is highly metabolically active compared to adults [17], providing an optimal environment for viral replication which often results in a high viremia. We have previously observed that immature DCs are efficient targets of SIV infection after vaginal challenge [22,31]. They can also play roles in HIV/SIV transmission in other models [35–37]. However, immature DCs are not a preferential target of the SHIV-1157ipd3N4 virus utilized in this neonatal model of oral viral exposure. For example, it has previously been shown that neonates have more proliferating CD4+ T cells in the small intestine than adult macaques and that these proliferating T cells are selectively infected after intravenous injection of SIVmac251 in neonates [14,15]. Similarly, our data show that T cells, specifically non-Th17 T cells, were the largest subset of infected immune cells in all tissue types analyzed after oral SHIV-1157ipd3N4 challenge (**Figs 4 and S1**). It has been shown that breastfed and bottle fed RMs develop distinct immune systems and gut microbiota, with a major difference being in the development of Th17 T cells [54]. Ardeshir et. al. demonstrate that breastfed RMs develop robust memory and Th17 T cell populations whereas bottle fed RMs did not [54]. This distinction is very important and could explain the results shown here that using this model of oral viral challenge in nursery reared RMs, Th17 T cells were not found to be the largely infected. However, it is important to note that we did not look further into what CD4+ T cell subset makes up our CCR6- CD3+ T cell population. Likewise, the CD3+ CCR6+ AG3+ cells that we observed throughout the oral mucosa and GI tracts of the neonatal RMs could be γδ T cells as they also express CCR6. While γδ T cells are a minor subset of T cells in both humans and RMs, they have been shown to be infected by HIV and SIV [55,56]. Experiments to identify particular subsets of T cells are infected in this model of oral viral challenge remain to be investigated in future experiments.

These data demonstrate that HIV virions can distribute throughout and penetrate the entire GI tract while remaining intact at a very rapid rate hours after oral viral exposure. These These findings were only made through use of our technologies of PET/CT coupled with a photoactivatable GFP-tagged HIV. Furthermore, we also show that the entire GI tract is susceptible to viral infection after oral viral exposure. Our data indicate that the small intestine has the potential to be a primary site for viral infection and that CD4+ T cells are the primary target cells 96 hours after viral exposure. Taken together, our results provide more insight to the mechanism behind acute viral infection in neonates after oral exposure.

## Experimental methods

### Ethics statement

PET/CT experiments were conducted at the New Iberia Research Center (NIRC) at the University of Louisiana at Lafayette. Studies examining LICh transduction and SHIV-1157ipd3N4 infection were conducted at the Southwest National Primate Research Center at the Texas Biomedical Research Institute (Texas Biomed), in San Antonio, Texas All procedures were approved by the Animal Care and Use Committees at Texas Biomed (IACUC: 1441 MM) and NIRC (IACUC: 2017-8791-002). All studies were performed in accordance with the recommendations in the Guide for the Care and Use of Laboratory Animals.

## Virus production

To generate PA-GFP-BaL, we co-transfected the R5-tropic, R9-BaL infectious molecular clone construct with a plasmid expressing a photoactivatable GFP (PA-GFP) [57] fusion with HIV VPR (PA-GFP-VPR) as previously described [28–30]. The replication competent virus labeled with PA-GFP-VPR generated by polyethylenimine transfection of human 293T cells in DMEM medium containing 10% heat-inactivated fetal calf serum, 100U/ml penicillin, 100μg/ml streptomycin, and 2mM l-glutamine. After 24 to 48 hours, virus was harvested, filtered at 0.45 μm and stored at -80˚C [28]. Viral particles were concentrated and enriched by centrifugation through a sucrose cushion.

LICh reporter virus was produced as previously described [31]. Briefly, a SIV-based pseudovirus vector system was generated from modifications of the SIV3 vector system [58]. The firefly luciferase gene is expressed through a poliovirus internal ribosome entry site (IRES) [59]. Transcription of both luciferase and mCherry are driven from the constitutive immediate-early CMV promoter and their expression is enhanced by WPRE for robust expression. LICh is produced by transfecting 293T cells with four plasmids: LICh reporter genome, SIV3+ packaging vector, REV expression plasmid DM121, and JRFL envelope. Viral supernatants were collected 48 hours post-transfection, purified through 0.45μm filters, concentrated over sucrose cushions, and resuspended in PBS. Concentrated virus was stored at −80˚C.

SHIV-1157ipd3N4 was generated as previously described [32]. Naïve RM peripheral blood mononuclear cells (PBMCs) were stimulated with concanavalin-A, followed by infection with SHIV-1157ipd3N4 that was harvested from 293T cells in the presence of human IL-2 (20U/mL) and TNF-α (10ng/mL). The PBMC-derived virus stock has a p27 concentration of 227ng/mL and $4x10^6$ $TCID_{50}$ per mL as titrated in TZM-bl cells.

## DOTA-labeling of virus

HIV virus was DOTA-labeled as previously described [26,30]. PA-GFP-BaL was labeled with a dodecane tetraacetic acid (DOTA) chelator, which allowed for attachment of $^{64}$Cu. Two buffers were prepared using a chelating resin to remove all free copper: 0.1 M sodium phosphate buffer (pH 7.3) and 0.1 M ammonium acetate buffer (pH 5.5). Chelex 100 Chelating Resin (5g, BioRad, Hercules, California) was added to 100 ml of each buffer, incubated with stirring for 1 hour at room temperature, and filtered at 0.22 μm for sterilization. Concentrated virus was resuspended in PBS and a 1:10 volume of 1 M sodium bicarbonate added. DOTA-NHS-ester (Macrocyclics, Dallas, Texas) was dissolved in the 0.1 M sodium phosphate buffer. The two solutions were combined (0.3mg DOTA-NHS-ester per 500ng of virus, as detected by p24 assay), and incubated on a rocker in the dark at room temperature. After 30 minutes, the buffer was exchanged for the 0.1 M ammonium acetate using a Zeba column 40K (Thermo Fisher Scientific, Waltham, Massachusetts), wash steps completed per manufacturer's protocol, and virus (PA-GFP-BaL-$^{64}$Cu) collected and frozen for shipment to New Iberia Research Center (NIRC) at the University of Louisiana at Lafayette.

## $^{64}$Cu labeling of virus particles

A solution of $^{64}$CuCl$_2$ (University of Wisconsin-Madison) was neutralized with Chelex-treated 1 M NH$_4$OAc (Sigma) to a pH of 5.5, and an aliquot (~185MBq) incubated with DOTA-PA-GFP-BaL stock at 37˚C for one hour. The sample was purified with a Zeba desalting spin column (30K MWCO, Thermo Fisher), eluted with PBS (Thermo Fisher), and labeling efficiency was evaluated. Labeled virus (10–37MBq) was mixed with unlabeled virus immediately prior to oral challenge.

## Non-human Primate Studies: oral viral challenge

In total, 14 Indian-origin rhesus macaques (*Macaca* mulatta) were used. All 14 animals used in the experiments described in this manuscript were nursery reared by veterinary staff at either Texas Biomed or NIRC. Four animals that were used for PET/CT experiments received 2mL of PA-GFP-BaL (1,000ng/mL) + 0.25mL of PA-GFP-BaL-$^{64}$Cu ((1,000ng/mL) in 2mL of Pedialyte for a total volume of a 4.25mL feeding (25). One animal was inoculated with a single dose (8mL) of LICh reporter virus and sacrificed at 96 hours to generate proof-of-principle data (**Fig 3**). One animal was inoculated with four doses of 2mL of LICh + 3.5mL SHIV-1157ipd3N4 (8mL of LICh and 14mL of SHIV-1157ipd3N4 in total) and sacrificed at 53 hours for a low dose, early time point challenge (**Table 1**). The remaining eight animals were challenged via eight feedings of 5mL SHIV-1157ipd3N4 over a span of two days (for a total of 40mL of SHIV115ipd3N4 over the course of the experiment) + one dose of 8mL LICh, which was included in the final feeding (**S1 Table**). Animals were sacrificed 96 hours after the initial viral challenge feeding. All animals were inoculated with virus that was mixed with Pedialyte via oral bottle feeding. Animals were humanely sacrificed with an overdose (100mg/kg) of pentobarbital while under isoflurane anesthesia (Euthasol, Virbac, Westlake, Texas) or telazol anesthesia. For all experimental conditions, the oral mucosa, entire gastrointestinal tract, spleen, liver, trachea, lungs, and cervical lymph nodes were removed. For PET/CT experiments, tissues were cut into 1-cm$^2$ pieces and frozen in optimal cutting temperature (OCT) media (Thermo Fisher Scientific). Once frozen tissue was no longer radioactive, it was shipped on dry ice to the Hope Lab at Northwestern University. For experiments examining LICh transduced and SHIV-1157ipd3N4 infected cells, samples were stored in RPMI after necropsy, and shipped on ice overnight to Northwestern University.

## NHP studies: imaging (PET, CT, and IVIS)

Animals were sedated with intramuscularly with 10mg/kg Telazol/ketamine (Zoeis, Parsippany-Troy Hills, New Jersey) and 1–3% isoflurane in 100% oxygen for the following scans. The animal's body was immobilized in dorsal recumbency in a vacuum-sealed veterinary positioner, and body temperature maintained with a warm air blanket (3M Bair hugger Model 505 warming unit, Saint Paul, Minnesota) and water-circulating heating pads. In addition, respiration, movement, and mucosal coloration (PET Scanner) were visually assessed. PET/CT scans were acquired using a Philips Gemini TF64 PET/CT scanner. The final CT image was compiled from 250 to 300 slices, depending on animal size. PET-CT combined images were analyzed using MIM software. Standard uptake values were measured using the volume regions of interest (ROI) tool and compared and normalized across animals. All scans lasted 20 minutes. Initial PET scans were obtained immediately following oral viral challenge. Second scans were performed one hour after oral challenge. PET scans also occurred at two and four hours for final *in vivo* tracking of radiolabeled virus. CT scans were performed immediately following the last PET scan. After sacrifice, PET images of whole tissues were taken. After tissue processing, each block was scanned and compared to a standard control for PET intensity.

Tissues were soaked in 100mM d-Luciferin (Biosynth) for a minimum of 10 minutes and placed in *In Vivo* Imaging System (IVIS) machine to examine luciferase activity (Fig 1B). Tissues positive for luciferase were cut into 1x1 mm$^2$ pieces and frozen in optimal cutting temperature (OCT) media for PCR and microscopy. For experiments with PA-GFP-BaL, four animals received 0.5mL $^{64}$Cu-PA-GFP-BaL and unlabeled 1mL of PA-GFP-BaL in Pedialyte. Animals were humanely sacrificed with an overdose of a solution containing pentobarbital (100mg/kg) while under isoflurane anesthesia. For all experimental conditions, the oral mucosa and GI tract were removed and separated by individual tissues, mesenteric lymph

nodes were separated from the colon and stored as individual lymph nodes. All tissues were frozen in optimal cutting temperature media (OCT) for microscopy.

## Nested PCR

Genomic DNA for nested PCR was isolated from frozen tissue sections embedded in OCT using the Qiagen DNeasy Blood and Tissue Kit as per the manufacturer's instructions. The initial nested PCR reactions were performed with 250ng of DNA per reaction and DreamTaq (Thermo Scientific). The second round of PCR was performed using 2ul of the first-round reaction products. For detection of LICh transduction, PCR was performed to identify mCherry DNA using the following primers: outside forward 5'-ACATGTGTTTAGTCGAGG-3', outside reverse 5'-CAGTCAATCTTTCACAAATTTTGTAATCC -3', inside forward 5'-CC GACTACTTGAAGCTGTCCTT-3', and inside reverse 5'-GTCTTGACCTCAGCGTCGTA GT-3'. For detection of SHIV-1157ipd3N4 infection, PCR was performed to identify gag DNA using the following primers: outside forward 5'-ATTAGCAGAAAGCCTGTTGGAG-3', outside reverse 5'-AGAGTGTCCTTCTTTCCCACAA-3', inside forward 5'-CATTCACGCAGA AGAGAAAGTG-3', inside reverse 5'-GGTATGGGGGTTCTGTTGTCTGT-3'. Each DNA sample was tested in 12 replicates. Sequences were confirmed by extracting DNA bands using Qaigen QIAquick Gel Extraction Kit and analysis with the second-round primers.

## Fluorescent microscopy and image analysis

For all imaging, twelve-to-fifteen-micron tissue sections were cut and fixed with 3.7% formaldehyde in PIPES buffer for 10 minutes at room temperature. To study distributions and localization of PA-GFP-BaL, nuclei were stained with Hoechst (1:25,000, ThermoFisher Scientific). Coverslips were mounted with DakoCytomation mounting medium (Burlington, Ontario, Canada) and sealed with nail polish. Twenty to twenty-two Z-stack images at 0.5-μm steps were obtained by deconvolution microscopy on a DeltaVision inverted microscope (GE, Boston, Massachusetts) at 100x. The 20–22 images were taken in an unbiased manner by following the linear path of the luminal surface of the mucosa with the distance between each image dictated by bleaching caused by photoactivation. Images were analyzed with softWoRx software (Applied Precision, Issaquah, Washington). Number of total virions and penetrating virions were counted. Virions were defined by existing in more than three Z planes and the post-activation signal had to be at least three times higher than the pre-activation signal. Penetrating virions were defined by being at least 1μm deep into the epithelium; all penetration depths were measured in microns.

Tissue sections used to investigate infected cells were then washed in PBS and tissue sections were blocked in donkey serum (10% normal donkey serum, 0.1% Triton-X-100, 0.01% NaN$_3$) for 30 minutes at room temperature; blocking solution was used for staining buffer throughout experiment. For mCherry detection, tissue sections were stained with primary antibodies for CD3 (one drop of the prediluted antibody per 200μl staining buffer, clone SP7, Abcam) and HIV envelope (1:300 AG3, AIDS Reagent Repository) for two hours at room temperature. Slides were washed with PBS and then stained with secondary antibodies donkey anti-rabbit-AF488 and donkey anti-mouse-AF647 (1:1000, Jackson ImmunoResearch, West Grove, Pennsylvania) for one hour at room temperature. Nuclei were stained with Hoechst (1:25,000) for 10 minutes at room temperature. Adjacent tissue sections were stained with a primary rabbit anti-firefly luciferase (1:200, Abcam) for two hours at room temperature. Slides were washed with PBS and then stained with secondary antibodies donkey anti-rabbit-AF488 (1:1000). Nuclei were stained for with Hoechst (1:25000). Coverslips were mounted with DakoCytomation mounting medium and sealed with nail polish. Images were taken in 3x5 Z-stack panels at 0.5μm steps for 30 steps by deconvolution microscopy on a DeltaVision

inverted microscope (GE) at 60x. Images were analyzed with softWoRx software (Applied Precision, Issaquah, Washington). Upon finding cells that appeared to be mCherry+, location of the cells was recorded, and sections were marked. Slides were then taken to an A1R-Spectral confocal microscope (Nikon, Tokyo, Japan) to analyze the emission spectra of the previously found cells compared to the known emission spectra of mCherry at 610nm. Images were analyzed with NIS Elements-C software (Nikon).

To phenotype of SHIV-1157ipd3N4 infected cells, slides were stained with primary antibodies for CD3 (one drop of the prediluted antibody per 200μl staining buffer, clone SP7, Abcam) and HIV envelope (1:300 AG3, AIDS Reagent Repository) for two hours at room temperature. Slides were washed with PBS and then stained with secondary antibodies donkey anti-rabbit-AF488 and donkey anti-mouse-AF594 (1:1000, Jackson ImmunoResearch) for one hour at room temperature. Tissue sections were then washed and stained with an AF647-directly conjugated antibody towards CCR6 (1:200, clone G034E3, BioLegend) at 37˚C for one hour. Nuclei were stained for with Hoechst (1:25000). Coverslips were mounted with DakoCytomation mounting medium and sealed with nail polish. Image panels containing 30 sections in the Z plane at 0.5um steps were taken and deconvoluted with softWoRx software on a DeltaVision inverted microscope. Five 40x images were taken for each sample. Each image consisted of a stitched panel of three 40x images by 5 40x images to include the epithelium. Infected cells were counted, cell phenotypes were identified (T cells–$CD3^+$, Th17 cells–$CD3^+$ $CCR6^+$, immature dendritic cells–$CD3^-$, $CCR6^+$, and other–$CD3^-$, $CCR6^-$), and cell subsets were recorded as parts of a whole (100%).

## Statistical analysis

All statistical analyses were performed using R version 4.0.2. To perform group comparisons per necropsy time, different mixed-effects models were fitted separately for the 2- and 4-hour necropsy animals. For each dataset, we used the best fitting model depending on the nature of the data analyzed. We always included animal as a random effect in the models. Virion counts per necropsy time were modeled using a negative binomial generalized mixed model to test for differences between anatomical locations, controlling for the number of images taken per animal and tissue. For this dataset we also tested zero-inflated negative binomial models due to the high number of zeros observed in some anatomical regions, but these models did not significantly improve the performance of the negative binomial model. To test for differences in number of penetrating virions present in the different anatomical locations per necropsy time, we transformed the penetrating depth data into a binary categorical variable by considering penetrating virions as those deeper than one micron from the epithelial surface data, while defining the rest as non-penetrating virions. We subsequently fitted a binomial generalized linear mixed-effects model including the anatomical location as the predictor variable, controlling for the number of images taken per tissue and animal. Finally, we tested for differences in penetrating depth among virions considered as penetrating by the previous definition between the different anatomical locations. We used a linear mixed effects model, also controlling for number of images. We performed all possible contrasts within each of the models, adjusting the false discovery rate for multiple comparisons using Benjamini-Hochberg Procedure. A false discovery rate (FDR) significance cut-off was set at FDR<0.05 for every comparison.

## Supporting information

**S1 Table. Identification of neonatal rhesus macaques used in LICh and SHIV-1157ipd3N4 studies.** *Day of harvest, date of harvest.
(PDF)

**S1 Fig. Phenotype of SHIV-1157ipd3N4 infected cells in neonatal RM after oral viral exposure.** Quantification of SHIV-1157ipd3N4 infected cells found in the GI tract of each animal examined in Figs 3 and 4. Graphs depict the percentage of infected cell types as parts of a whole in each individual animal identified by fluorescent microscopy. Infected cell types were categorized as five cell types: T cells (CD3+), TH17 T cells (CD3+, CCR6+), CCR6- CD3+ T cells, Immature DCs (CD3-, CCR6+), Other (CD3-, CCR6-). Total cell counts were taken in five 40x panels in every animal.
(TIF)

## Acknowledgments

Thanks to the research and animal care teams at Texas Biomed and NIRC. Thank you to Danijela Maric and Katarina Kotnik Halavaty for instruction on experimental techniques.

## Author Contributions

**Conceptualization:** Ruth M. Ruprecht, Thomas J. Hope.

**Data curation:** Roslyn A. Taylor, Michael D. McRaven, Ann M. Carias, Meegan R. Anderson, Edgar Matias, Mariluz Araínga, Edward J. Allen, Yanique Thomas, Francois J. Villinger.

**Formal analysis:** Roslyn A. Taylor, Ramon Lorenzo-Redondo, Yanique Thomas, Thomas J. Hope.

**Funding acquisition:** Roslyn A. Taylor, Ann M. Carias, Ruth M. Ruprecht, Thomas J. Hope.

**Investigation:** Roslyn A. Taylor, Michael D. McRaven, Ann M. Carias, Mariluz Araínga, Thomas J. Hope.

**Methodology:** Roslyn A. Taylor, Michael D. McRaven, Ann M. Carias, Meegan R. Anderson, Edgar Matias, Mariluz Araínga, Edward J. Allen, Kenneth A. Rogers, Sandeep Gupta, Viraj Kulkarni, Samir Lakhashe, Amanda Strickland, Francois J. Villinger.

**Project administration:** Ruth M. Ruprecht, Thomas J. Hope.

**Resources:** Francois J. Villinger, Ruth M. Ruprecht, Thomas J. Hope.

**Supervision:** Francois J. Villinger, Ruth M. Ruprecht, Thomas J. Hope.

**Validation:** Roslyn A. Taylor, Michael D. McRaven, Thomas J. Hope.

**Visualization:** Roslyn A. Taylor, Ruth M. Ruprecht, Thomas J. Hope.

**Writing – original draft:** Roslyn A. Taylor.

**Writing – review & editing:** Roslyn A. Taylor, Michael D. McRaven, Ann M. Carias, Meegan R. Anderson, Edgar Matias, Mariluz Araínga, Edward J. Allen, Kenneth A. Rogers, Sandeep Gupta, Viraj Kulkarni, Samir Lakhashe, Ramon Lorenzo-Redondo, Yanique Thomas, Amanda Strickland, Francois J. Villinger, Ruth M. Ruprecht, Thomas J. Hope.

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
