## [Decision Letter · Decision Letter 0]

21 Aug 2021

Dear Professor Hope,

Thank you very much for submitting your manuscript "Localization of infection in neonatal rhesus macaques after oral viral challenge" for consideration at PLOS Pathogens. As with all papers reviewed by the journal, your manuscript was reviewed by members of the editorial board and by several independent reviewers. In light of the reviews (below this email), we would like to invite the resubmission of a significantly-revised version that takes into account the reviewers' comments.

We cannot make any decision about publication until we have seen the revised manuscript and your response to the reviewers' comments. Your revised manuscript is also likely to be sent to reviewers for further evaluation.

Sincerely,

David T. Evans

Associate Editor

PLOS Pathogens

Susan Ross

Section Editor

PLOS Pathogens

Kasturi Haldar

Editor-in-Chief

PLOS Pathogens

orcid.org/0000-0001-5065-158X

Michael Malim

Editor-in-Chief

PLOS Pathogens

orcid.org/0000-0002-7699-2064

Reviewer's Responses to Questions

**Part I - Summary**

Reviewer #1: This manuscript reveals novel information on virion trafficking and initial infected cell types in the infant gut after oral virus challenge in bottlefed infant monkeys. This is important work that may aid in developing interventions for this most common route of HIV infection in infants that is currently not addressed by antiretroviral-based prevention alone. Yet, there are limitations in the model that should be further discussed and indicated as limitations, such as the impact of breast milk itself on the virion trafficking and infection of cells in the mucosa.

Reviewer #2: HIV-1 infection by breastfeeding remains a major cause of new pediatric infections. Defining the kinetics of virus dissemination and identifying early sites of virus replication may inform novel prevention strategies. The study by Taylor et al. utilized the neonatal rhesus macaque model to address these questions. Applying PET/CT and fluorescent microscopy, the investigators can demonstrate that virus rapidly disseminate from the oral cavity to the rectum within 4 hours of virus feeding. By 96 hours, virus can be detected throughout the entire gastrointestinal tract, with the majority of infected cells representing non-Th17 T cells in the small and large intestine.

The strength of the study lies in capturing of very early (4 hours) viral foci at sites distal form the virus exposure site by applying PET/CT in combination with fluorescent deconvolution microscopy. The in vivo analysis of virus dissemination is not an easy endeavor. The thoroughness of the evaluation of tissue sections from the entire GI tract to identify the phenotype of virally infected cells is acknowledged. However, the finding that T cells represent the main infected cell type at 96 hours is not necessarily novel (Wang at al, 2010, Blood, 116:4168; Amedee et al., 2018, AIDS Res Hu Retroviruses, 34:286), even in the light of different routes of virus exposure. The fact that Th17 CD4+ T cells did not represent the majority of infected T cells is interesting and should be discussed in the context of animal rearing. Ardeshir et al. demonstrated previously that the establishment of Th17 cells is more pronounced in breastfed vs. nursery-reared infant rhesus macaques (Ardeshir et al., 2014, Sci Transl. Med., 6:252ra120).

Reviewer #3: This manuscript by Taylor et al utilizes recent technological advancements to address the question of how the SIV/SHIV virus infects an infant macaque monkey following oral exposure in the form of formula feeding. There are a number of findings in the paper that validate previous publications, as well as to provide additional insights into what is occurring at times following oral exposure of virus or viral reporters to infant macaques. Overall, while the technology utilized does provide interesting insights, there are a number of major issues that dampen the overall enthusiasm.

**Part II – Major Issues: Key Experiments Required for Acceptance**

Reviewer #1: 1. while these studies are very revealing in the trafficking of virions and the initial infected cells, it may not represent the low dose exposure of breast milk feeding, as only 10% of breastfeeding infants of HIV+, untreated moms will become infected over the entire period of breastfeeding. The differences of this model and physiologic breastfeeding should be discussed, in particular, why so few infants become infected over the period of breastfeeding even in the absence of antivirals yet this model indicated that all challenged infants had virus-infected CD4+ T cells in the GI mucosa

2. The statistics need to be reviewed and further described, as I am not sure how an FDR-corrected significant p value can be achieved with only 4 animals using paired tests (since the observations come from the same animal). The multiple images reviewed should be averaged for each infant as opposed to used as multiple data points.

3. Tables 2 and 3 should indicate magnitude of infected cells identified in each tissue, even though it seems the technologies are quantifiable. And perhaps as a heat map instead of a table would be a better visual.

4. As the LICh experiment had many technical challenges presented in the results and few reliable data points, the data from this approach should be removed, and this employed approach can mainly be included in the methods and discussion

5. The method of bottle-fed inoculation includes pedialyte as the diluent, which is not a physiologic representation of breast milk or even formula. The authors should discuss the limitations of this approach and how the virion distribution and infections may have changed with the virus diluted in breast milk, which has a number of innate anti-microbial proteins, lipids, as well as maternal antibodies.

Reviewer #2: 1. Although the authors state that mesenteric lymph nodes were only collected from two animals, what was the rationale of not including lymph nodes in this analysis?

2. Tissue sections were stained with CD3 and CCR6 to distinguish between T cells, Th17 cells and immature DC. The addition of a CD4 antibody would have allowed to quantitate the number of CCR6+CD4+ T cells (likely Th17+) of total CD4+ T cells which might more accurately reflect CD4+ Th17 cells as CCR6 is also expressed by �� T cells and certain CD8+ T cells. Along the same lines, the statement that the results of the study provide insight into mechanisms (line 441) of acute infection after oral SHIV exposure is not justified and would warrant a more detailed analysis of various CD4 populations and their viral coreceptors

3. The timing of feeding in the eight animals that were infected with SHIV1157ipd3N4 needs to be clarified. The Method section lists that infants were few eight times with 5 ml of the virus. It is important to know whether the feeding interval. If the feeding was spread out over multiple times, it is feasible that the 96-hour time point could refer to a few hours post the last virus exposure up to 96 hours depending on what exposure resulted in infection.

4. Please add a statement to the rearing of the neonatal macaques, presumably nursery-reared? In addition, please also list the age of the 4 animals used for PET/CT studies and for the two animals in the LICh only studies.

5. Line 348: CCR5 is certainly a coreceptor for HIV, but CCR5 is not commonly referred to as a mucosal homing marker. Although activated (and CCR5+) T cells are more commonly found at mucosal sites, the integrin alpha4beta7 would be a more appropriate mucosal homing marker for intestinal tissues.

Reviewer #3: Major issues:

1. There is a concern with regard to using radioactive copper as a way to identify entry points of SIV/SHIV in orally inoculated infant macaques. Looking at the images in Figures 1 and 2, it appears that the virus with radioactive copper simply followed the liquid down the digestive tract unless it hit an air bubble (or went to the large intestine). I think this says that the virus can potentially make it down that far in the amount of time assessed, which is informative. However, the authors should be careful not to over interpret the location of the radioactive copper or the GFP as indicative of the most likely entry points since the entire digestive tract appears to be bathed in it.

2. Figure 2 E, F and G are clearly impactful and potentially important figures in this manuscript. However, there are a few issues that are concerning. First, the data is dependent on photo micrograph from Figure 1C being clear and easy to interpret, however only one GFP positive virus is easily discernable and that is in the inset. Without clearer pictures from different tissues it is difficult to know how to interpret data from graphs. Second, the y axis for Figures 2E and 2F are easy to understand. Although a brief description is present in methods for total virions counted, this is not sufficient for understanding how the different tissues were evaluated. For example, why do the methods say 20 or 22 images, shouldn’t that have been the same number for each tissue if you were counting total numbers of GFP positive spots? The exact area being evaluated would also be good to know. Third, the differences from 2 to 4 hours resembles what one would expect for a virus particle that was traveling with formula. It is not clear that this says very much about viral entry, I think it says something about potential for viral entry.

3. Figure 3. The LICh vector analysis was problematic for reasons raised by authors. Furthermore, there does not appear to add anything substantial to conclusions from the manuscript. It seems that all figures associated with this can be removed without impacting any of the conclusions from the manuscript. Use of vector can still be mentioned, but in my opinion a clear explanation of why these data are included is needed if they are to remain.

4. CCR6. Microscopy data for CCR6 is not convincing. Therefore, conclusions regarding infection of the different cell subsets is called into question. Can better micrographs be utilized instead of what is included here or were the positive cells just slightly above the background?

5. Detection of SHIV infected cells in the intestines after 4 days might reflect more rapid replication here in lymphoid tissues associated with gut mucosa. Therefore, you may not be looking at entry, but rather a viral replication site that is replicating a viral population that entered at a different site. This is because 4 days is quite a long time from when the macaque was infected. Is there any data from earlier time points that could help to sort this out? Also, and this applies to comment 2 above as well, it would be great to have more info in figure legends regarding what it means when the data reports on ‘number of cells’.

6. Subtype C SHIV. It is not clear why the authors did not simply use a SIV for this study. Transmission data with a SHIV might differ from a virus that naturally infects nonhuman primates. A better explanation with regard to why this virus was utilized is warranted.

7. Discussed throughout the manuscript as well as at end of discussion (Line 438-440): From the paper “Furthermore,we also show that the entire GI tract is susceptible to viral infection after oral viral exposure. Our data indicate that the small intestine is the primary site for viral infection and that CD4+ T cells are the primary target cells 96 hours after viral exposure.” The conclusion that the small intestine is the primary transmission point is difficult to conclude from the data provided. It seems since these findings provide some support, but some divergence, from previous studies (including Amadee et al 2018) the conclusions should reflect this. Also, data needs to be clearer and cleaner if the authors are to conclude that infection is occurring through the large intestines. As stated above, most of the data supporting this in the manuscript can also be explained as the virus or viral reporter simply being moved through the digestive tract with the formula. Suggest altering this conclusion.

8. The title is not accurately describing the findings presented in the manuscript. One suggestion for a new title might be: Detection of SHIV and viral reporters within the gastrointestinal tract following oral infection of neonatal rhesus macaques

**Part III – Minor Issues: Editorial and Data Presentation Modifications**

Reviewer #1: 1. with such a small number of infants included in each necropsy study, etc - each data point on virion frequency, etc should be reported per animal, not averaged

2. authors should address the potential limitation with the dual infections of competition between the 2 viruses

3. in Discussion, include a reference for the paucity of CCR5+ CD4+ T cells in tonsils

4. The handwritting on images is difficult to read - can the image include text boxes for indicating the anatomic locations?

Reviewer #2: 1. In the Abstract, the first sentence is missing some words.

2. Figure 1: Please explain what area of the picture the insert in Panel C refers to.

3. Figure 2, Panel E: The mean values are barely readable.

Figure2, Panel F: Please clarify that the frequencies refer to total frequencies of virions counted at the specific anatomic site over the 2- and 4-hour period combined. The Panel could be deleted because the data represent the results of Panel E in a different format.

4. Figure 3: The red background signal of Cell 1 in Panel A is not visible to the reviewer. Similarly, the colors in Panel C are difficult to distinguish. However, it is possible that the figures were not reproduced at print quality for the review stage.

5. Please check the reference section, most journal abbreviations are not listed correctly.

6. It is probably not necessary to list the initials of the individuals who prepared the virus stock, but the decision lies with the authors.

Reviewer #3: Minor comments:

• Line 179: Please replace “neck lymph nodes” with anatomically accurate definitions and replace (possibly cervical lymph nodes)

• Figure 1C is not labelled with a ‘C’

• Line 423, and anywhere prior: “other T cells” should be more accurately referred to as CCR6- CD3+ T cells

PLOS authors have the option to publish the peer review history of their article (what does this mean?). If published, this will include your full peer review and any attached files.

Reviewer #1: **Yes: **Sallie Permar, MD, PhD

Reviewer #2: No

Reviewer #3: No
---

## [Decision Letter · Decision Letter 1]

6 Nov 2021

Dear Professor Hope,

We are pleased to inform you that your manuscript 'Localization of infection in neonatal rhesus macaques after oral viral challenge' has been provisionally accepted for publication in PLOS Pathogens.

Best regards,

David T. Evans

Associate Editor

PLOS Pathogens

Susan Ross

Section Editor

PLOS Pathogens

Kasturi Haldar

Editor-in-Chief

PLOS Pathogens

orcid.org/0000-0001-5065-158X

Michael Malim

Editor-in-Chief

PLOS Pathogens

orcid.org/0000-0002-7699-2064

Reviewer Comments (if any, and for reference):

Reviewer's Responses to Questions

**Part I - Summary**

Reviewer #3: (No Response)

**Part II – Major Issues: Key Experiments Required for Acceptance**

Reviewer #3: (No Response)

**Part III – Minor Issues: Editorial and Data Presentation Modifications**

Reviewer #3: (No Response)

PLOS authors have the option to publish the peer review history of their article (what does this mean?). If published, this will include your full peer review and any attached files.

Reviewer #3: No

---

## [Editor Report · Acceptance letter]

12 Nov 2021

Dear Professor Hope,

We are delighted to inform you that your manuscript, "Localization of infection in neonatal rhesus macaques after oral viral challenge," has been formally accepted for publication in PLOS Pathogens.

Best regards,

Kasturi Haldar

Editor-in-Chief

PLOS Pathogens

orcid.org/0000-0001-5065-158X

Michael Malim

Editor-in-Chief

PLOS Pathogens

orcid.org/0000-0002-7699-2064